# Are Bert Family Good Instruction Followers? A Study on Their Potential And Limitations

**Yisheng Xiao** [1], **Juntao Li** [1][*], **Zechen Sun** [1], **Zechang Li** [2], **Qingrong Xia** [2], **Xinyu Duan** [2], **Zhefeng Wang** [2], **Min Zhang** [1]

[1] Soochow University, China        [2] Huawei Cloud, China

{ysxiaoo,zcsuns}@stu.suda.edu.cn, {ljt,minzhang}@suda.edu.cn
{lizechang1,xiaqingrong,duanxinyu,wangzhefeng}@huawei.com

## Abstract

Language modeling at scale has proven very effective and brought unprecedented success to natural language models. Many typical representatives, especially decoder-only models, e.g., BLOOM and LLaMA, and encoder-decoder models, e.g., Flan-T5 and AlexaTM, have exhibited incredible instruction-following capabilities while keeping strong task completion ability. These large language models can achieve superior performance in various tasks and even yield emergent capabilities, e.g., reasoning and universal generalization. Though the above two paradigms are mainstream and well explored, the potential of the BERT family, which are encoder-only based models and have ever been one of the most representative pre-trained models, also deserves attention, at least should be discussed. In this work, we adopt XML-R to explore the effectiveness of the BERT family for instruction following and zero-shot learning. We first design a simple yet effective strategy to utilize the encoder-only models for generation tasks and then conduct multi-task instruction tuning. Experimental results demonstrate that our fine-tuned model, Instruct-XMLR, outperforms Bloomz on all evaluation tasks and achieves comparable performance with mT0 on most tasks. Instruct-XMLR also possesses strong task and language generalization abilities, indicating that Instruct-XMLR can also serve as a good instruction follower and zero-shot learner. Besides, Instruct-XMLR can accelerate decoding due to its non-autoregressive generation manner, achieving around 3 times speedup compared with current autoregressive large language models. Although we also witnessed several limitations through our experiments, such as the performance decline in long-generation tasks and the shortcoming of length prediction, Instruct-XMLR can still become a good member of the family of current large language models.[1]

## 1 Introduction

Making large language models (LLMs) as good instruction followers further unleashes their power for different usages. Through instruction tuning with annotated examples and human preference feedback (if possible), the output of these LLMs can align with human expectations well on various tasks, e.g., understanding, generation, commonsense reasoning, under different settings, including zero-shot, few-shot, and chain-of-thought (Ouyang et al., 2022; Longpre et al., 2023). Many recent works have pointed out the key factors and insights into the success of instruction following techniques on the popular decoder-only (e.g., GPT (Brown et al., 2020), LLaMA (Touvron et al., 2023)) and encoder-decoder frameworks (e.g., T5 (Raffel et al., 2020a)), including but not limited to annotation quality (Zhou et al., 2023), data formats, and scaling (Chung et al., 2022). It can be predicted that more powerful alignment methods will occur for more model variants.

To enlarge the application range of alignment techniques, we alternatively explore the potential and limitations of the BERT family as instruction followers. To our knowledge, this is the first work that attempts to provide suggestions for building instruction-following models on the BERT family.

---

[*]Juntao Li is the corresponding author.
[1]https://github.com/LitterBrother-Xiao/Instruct_XMLR/tree/main

Our study is motivated by existing observations that: 1) the BERT family can also be zero-shot learners (Zhao et al., 2022; Wang et al., 2023). Further equipping them with instruction understanding capabilities will expand their usage scenarios; 2) their masking language model objective can support open-ended long text generation and meanwhile achieve much faster decoding speed than the popular autoregressive fashion (Liang et al., 2023b); 3) complicated real-world applications usually involves the collaboration of generative LLMs and expert models (Liang et al., 2023c; Shen et al., 2023), while the BERT family have moved countless tasks forward before the rising of large generative models and still hold the records for various downstream tasks.

**Contributions**  Throughout this paper, we mainly (1) provide a basic method that can hint at the potential and limitations of the BERT family as instruction followers; (2) present the key factors in making the BERT family competitive with the popular encoder-decoder and decoder-only models that have similar model scale and pretraining data; (3) disclose the risks and limitations (4) along with the possible reasons and solutions to give suggestions for further performance improvement.

**Important Observations**  Although this work is at the preliminary stage, we still have some important observations for reference and future research. First, the BERT family has the possibility to be good instruction followers that can generalize well across tasks and languages compared to the other popular LLMs with similar model sizes and data. Besides, our presented method is scalable, in which one can use more computation resources and data to improve performance significantly. Some of the problems are possibly owing to the limited capabilities of the backbone models since there are no previous BERT family models that are as powerful as the decoder-only and encoder-decoder counterparts yet. The inherent shortages of non-autoregressive decoding also result in unconvinced and unreliability to feed different tasks and scenarios.

In the rest of this paper, we first provide the background of text generation via the BERT family and instruction tuning in Section 2. We then present the details for post-training the BERT family as a language generator in Section 3 and detailed experimental settings for instruction fine-tuning in Section 4. Section 5 and 6 analyze their potential and the possible reasons behind the limitations.

## 2 RELATED WORK

### 2.1 TEXT GENERATION VIA BERT FAMILY

BERT (Devlin et al., 2019) revolutionized the field of Natural Language Processing by leveraging the Transformer architecture and large-scale pre-training. BERT is a typical encoder-only architecture consisting of a multi-layer bidirectional Transformer encoder (Vaswani et al., 2017) with stacked identical blocks. Building upon the foundation laid by BERT, researchers have further explored and expanded the capabilities of the original BERT architecture, leading to the development of the BERT family, such as RoBERTa (Liu et al., 2019), ELECTRA (Clark et al., 2020), DeBERTa (He et al., 2020), and XLM-R (Conneau et al., 2020), among others. The bidirectional modeling characteristic and parallelizable masked language model (MLM) objective enables it to learn contextual word representations, facilitating the capture of comprehensive semantic information. Thus, the BERT family is renowned for its ability in natural language understanding (NLU) tasks. However, there is a scarcity of research that delves into their potential for text-generation tasks. Previous works (Dong et al., 2019; Wang & Cho, 2019) have theoretically indicated that the BERT family can generate coherent and high-quality textual content. However, the primary usage of the BERT family has been focused on extracting contextual features rather than generating text by itself (Zhu et al., 2019; Guo et al., 2020; Yang et al., 2020). Recent works (Chan & Fan, 2019; Jiang et al., 2021; Su et al., 2021; Liang et al., 2023b;a) explore the application of the BERT family in non-autoregressive generation using MLMs and receive positive feedback regarding performance. Their settings still follow the traditional pre-training and task-specific fine-tuning paradigm. In our work, we further explore the potential of becoming multitask instruction followers, which has been well explored in recent autoregressive language models but is a blank area for BERT family.

## 2.2 INSTRUCTION TUNING

Instruction tuning refers to the process of fine-tuning LLMs on an instruction dataset consisting of (*instruction*, *output*) pairs in a supervised fashion, which narrows the gap between the next-word prediction objective of LLMs and the users' objective of having LLMs adhere to human instructions (Zhang et al., 2023). The concept of merging diverse NLP tasks as generative tasks was pioneered by the T5 model (Raffel et al., 2020b). This method effectively simplifies the application of LLMs and paves the way for further advancements in instruction tuning models. Subsequent instruction tuning models like FLAN (Wei et al., 2021; Chung et al., 2022)(Wei et al., 2021; Chung et al., 2022) and T0 (Sanh et al., 2021) have further improved performance across diverse tasks by incorporating more task-specific instructions during the pre-training phase. Currently, instruction tuning represents an important research direction in NLP. The open-source community offers a variety of instruction datasets, such as xP3 (Muennighoff et al., 2022), Alpaca (Taori et al., 2023), and Dolly (Conover et al., 2023), as well as instruction fine-tuned LLMs, such as BLOOMZ (Muennighoff et al., 2022), FLAN-T5 (Chung et al., 2022), and Alpaca (Taori et al., 2023). However, the backbone of present instruction fine-tuned LLMs is mainly encoder-decoder and decoder-only based. The instruction-following capabilities of the BERT family, which are encoder-only based models, are severely under-explored. In this work, we introduced Instruct-XMLR to explore the potential and limitations of the BERT family for instruction following.

## 3 METHODOLOGY

### 3.1 UTILIZING BERT FAMILY FOR LANGUAGE GENERATION

Dramatically different from the prevalent left-to-right autoregressive unidirectional language models, the conditional independent factorization due to the bi-directional nature of BERT family captures more complex dependencies between tokens during training and also results in difficulty in generating reliable texts from scratch. In this section, we review the potential solution to utilize BERT family for language generation. As shown in previous work (Wang & Cho, 2019; He et al., 2022), BERT can be viewed as a Markov random field language model and then sample texts. Specifically, the sequence $Y = (Y_1, Y_2, ..., Y_T)$ can be viewed as random variables of an undirected fully-connected graph $S$, and this full graph can be defined as a clique in the Markov Random Field. Then the potential of this graph clique can be decomposes into a sum of $T$ `log`-potential terms :

$$\psi(S) = \prod_{i=1}^{T} \psi_i(S) = \exp\left\{\sum_{i=1}^{T} \log \psi_i(S)\right\}, \tag{1}$$

where $S$ denotes the fully-connected graph created from the original sequence $Y$, $T$ denotes the number of random variables, i.e., the length of sequence $Y$. This can also be simplified by Conditional Distribution and Pseudo Log-Likelihood which maximizes the probability of each variable rather than joint probability of the entire graph:

$$\psi(S) = \frac{1}{T} \sum_{i=1}^{T} \log \mathcal{P}(y_i | S_{\setminus i}). \tag{2}$$

Then for BERT family, $S_{\setminus i}$ can represent that $y_i$ is replaced by the masked token [MASK] in the entire sequence $Y$, $y_i$ can represent each masked token. Then, the original training objective can be approximated as a masked language model (MLM), where each potential $\psi(S)$ is based on the model probability $\mathcal{P}(y_i | S_{\setminus i}, \theta)$. Given the sequence $Y$, we can achieve:

$$\psi(Y) = \frac{1}{T} \sum_{i=1}^{T} \log \mathcal{P}(y_i | Y_{\setminus i}) \approx \frac{1}{R} \sum_{i=1}^{R} \log \mathcal{P}(y_i | Y_{\setminus i}, \theta), \tag{3}$$

where $R$ is the number of masked tokens sampled from $Y$, and the upper bound is the length of $Y$. Then we can expand this non-conditional language model to conditional scenarios, i.e., sequence to sequence generation. Given a training pair $(X, Y)$, the model maximizes the following likelihood:

$$\psi(Y) = \frac{1}{R} \sum_{i=1}^{R} \log \mathcal{P}(y_i | Y_{\setminus i}, X, \theta). \tag{4}$$

During inference, we begin with a fully $Y$ with all tokens masked and generate them in each step of $|Y|$ (Wang & Cho, 2019). Notice that the conditional MLM and Mask-Predict are suitable for BERT family, which keeps the training and inference objective consistent with pre-training.

## 3.2 ONE-STEP TRAINING WITH DYNAMIC MIXED ATTENTION

We can draw inspiration from the previous practices for language generation; the encoder-decoder-based models feed the source sequence $X$ into the encoder to extract the representations and use the decoder to learn the internal relationships of the target sequence $Y$. Then, a cross-attention module is inserted into each decoder layer to aggregate the source representation and the target sequence. The key factor to language generation is how to learn the conditional probabilities $\mathcal{P}(Y|X)$ well, and this is the same for BERT family. However, unlike previous encoder-decoder-based models, the BERT family only contains a single multi-layer bidirectional Transformer encoder. To best match the traditional learning formats, we first build an additional model to perform like the decoder, then introduce mixed attention module (He et al., 2018; Liang et al., 2023b) to substitute for the original cross-attention and self-attention modules in the decoder. Specifically, given a training pair $(X, Y)$, the pre-training MLM containing $L$ layers where each comprises one self-attention layer and one feed-forward layer. We can get the source representation $\mathcal{H}_{src}^l$ of each encoder layer $l$:

$$\hat{\mathcal{H}}_{src}^l = \texttt{Self\_Attention}(\mathcal{H}_{src}^{l-1}) + \mathcal{H}_{src}^{l-1}, \quad \mathcal{H}_{src}^l = \texttt{FFN}(\hat{\mathcal{H}}_{src}^l) + \hat{\mathcal{H}}_{src}^l. \quad (5)$$

For target sequence $Y$, we randomly select partial tokens to be replaced by the [MASK] token and feed this corrupted sequence into the additional MLM. Then, this model tries to recover these masked tokens during training, and get the target representation $\mathcal{H}_{tgt}^l$ of each decoder layer $l$:

$$\hat{\mathcal{H}}_{tgt}^l = \texttt{Mixed\_Attention}(\mathcal{H}_{src}^L \oplus \mathcal{H}_{tgt}^{l-1}) + \mathcal{H}_{tgt}^{l-1}, \quad \mathcal{H}_{tgt}^l = \texttt{FFN}(\hat{\mathcal{H}}_{tgt}^l) + \hat{\mathcal{H}}_{tgt}^l, \quad (6)$$

where $\oplus$ denotes the concatenation operation. Mixed-attention brings no additional parameters but just take the concatenated vector of source and target hidden states as key and value in the original attention mechanism. As a result, this additional MLM can completely share parameters with the MLM encoder. Thus, we can adopt only one pre-training BERT model to save model parameters and accelerate the training process. However, the mix-attention mechanism first acquires the last layer's source representation. We must pass the model twice during training, leading to low training efficiency. As a result, we introduce dynamic mixed-attention, which allows the model to get source representation and learn to predict the masked tokens at the same pass as the model. This mechanism adopts each source representation of the corresponding previous layer rather the last layer as:

$$\hat{\mathcal{H}}_{tgt}^l = \texttt{Mixed\_Attention}(\mathcal{H}_{src}^{l-1} \oplus \mathcal{H}_{tgt}^{l-1}) + \mathcal{H}_{tgt}^{l-1}, \quad \mathcal{H}_{tgt}^l = \texttt{FFN}(\hat{\mathcal{H}}_{tgt}^l) + \hat{\mathcal{H}}_{tgt}^l. \quad (7)$$

In practice, we can simply concatenate the source and the masked target sequence, then feed them into the model for training. Besides, as shown in Figure 1, we prevents the query of each source token attending the target sequence in the attention module, which keeps consistent with the inference process since there is no target sequence in advance. Dynamic mixed-attention is more matching with the pre-training tasks of BERT family, making the idea of shared parameters more reliable.

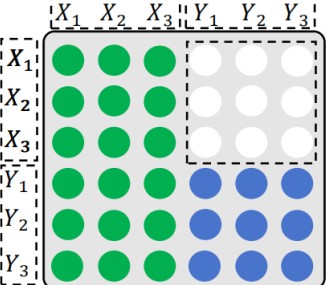

Figure 1: The overview attention masking, Green and Blue ones denote which can be attended while the white can not be attended.

## 3.3 TRAINING AND INFERENCE PROCESS

To better adapt to generation tasks, given a training pair $(X, Y)$, we uniformly mask 1 to $L$ (target length) tokens following CMLM (Ghazvininejad et al., 2019) rather the original fixed masking in BERT family. Besides, we also randomly mask a small ratio of tokens in source sequence to improve generality as mentioned in AMOM (Xiao et al., 2023). Then the training objective is to maximum the conditional MLM loss like the pre-training process:

$$\mathcal{L}_{\text{MLM}} = \sum_{i=1}^{M} \log \mathcal{P}(y_i|Y_{\text{M}}, X_{\text{M}}, \theta), \quad (8)$$

where $M$ is the number of masked tokens in masked target sequence $Y_M$. Notice the model do not need to predict the masked tokens in the source sequence $X_M$.

During inference, we adopt the same Mask-Predict algorithm in CMLM (Ghazvininejad et al., 2019), which adopts multiple iterations to generate the final sequence. Specifically, given the total decoding iteration $T$ in advance, we begin with the fully masked target sequence at the first decoding iteration. In the following $T-1$ iterations, a specific number of low-confidence tokens will be masked and re-generated. The number of masked tokens in each iteration can be computed as $n = \frac{T-t}{T} * L_Y$, where $t$ denotes the current iteration number. The model selects the specific masked tokens in the next iteration based on the prediction probability, where tokens with the lowest probability will be masked, and their scores will be updated after the new prediction. Besides, unlike traditional left-to-right auto-regressive models, we should obtain the target length before initializing the fully masked target sequence. Naturally, we can directly give a length before inference. We also introduce a length prediction module following previous non-autoregressive models where a special token `[LENGTH]` is used to predict the target length.

## 4 EXPERIMENTAL SETTINGS

In this section, we introduce our experimental settings in fine-tuning and evaluation process.

### 4.1 FINE-TUNING DETAILS

**Backbone Model.** Recent generative LLMs have been proven to be good instruction followers (Wei et al., 2021; Taori et al., 2023), especially for these with relatively large size (Chung et al., 2022; OpenAI, 2023). To fully explore this capability of BERT family, we also select our backbone model with large size version. As a result, we choose XML-R (Conneau et al., 2020), which is pre-trained on around one hundred languages with masked language modeling objective and has two large versions, XML-R$_{XL}$ and XML-R$_{XXL}$ (Goyal et al., 2021), containing 3.5 and 10.7 billion parameters respectively. Due to the cost of computing resources, We choose XML-R$_{XL}$ in our main experiments.

**Instruction Data.** For fine-tuning instruction data, to make full use of the multilingual knowledge during the pre-training process of XML-R, we select a multilingual instruction dataset xP3 (Muennighoff et al., 2022). xP3 adds 30 new multilingual datasets with English prompts and serves as the multilingual version of P3 (Sanh et al., 2021). Overall, xP3 contains 46 languages and a similar language distribution of ROOTS (Laurençon et al., 2022). In the original xP3 paper (Muennighoff et al., 2022), the authors use around 80 million samples for training, which is much more than the actual needs and increases the training expenses. As a result, we sample partial instruction samples (around 1/80) from the original training corpus in our experiment. Our sampled data only differs from the original data in sample size and remains strictly consistent in all other aspects, such as the composition ratio of each language, the number of tasks, prompt formats, etc.

**Implementation Details.** Our experiments use the pre-trained XML-R$_{XL}$, which contains 36 layers, 2560 hidden sizes, and 10240 feed-forward filter sizes. We implement all the experiments based on the Fairseq (Ott et al., 2019) library on 8 NVIDIA A100-PCIE-40GB GPU cards. We adopt the Adam (Kingma & Ba, 2014) as an optimization algorithm during training. The learning rate will warm up to 2e-5 in the first 500 updates and then decrease gradually with the `polynomial decay` schedule. We select the final model based on the validation performance.

### 4.2 EVALUATION PROCESS

**Baseline Models.** To make a relatively fair comparison between our fine-tuned model and other LLMs, we choose mT0-3.7B and BLOOMZ-3B since they have comparable model parameters and are also fine-tuned on the instruction dataset xP3. We give more comparison in Table 1. These models are with different and covers the current popular architectures. Need to notice a big difference exists in the number of training tokens, which always plays an important role on performance. Instruct-XMLR$_{XL}$ is trained on a bit more tokens than BLOOMZ-3B and half of mT0-3.7B during pre-training. While in fine-tuning Instruct-XMLR$_{XL}$ is training on much fewer tokens.

| Model | Instruction Tuning stage | | | | Pre-training Stage | | | |
|---|---|---|---|---|---|---|---|---|
| | Backbone | Architecture | Parameter | Tok. | Datasets | Size | Tok. | Len. |
| **BLOOMZ-3B** | Bloom-3b | Decoder-only | 3B | 15B | ROOTS | 1.6T | 366B | 2048 |
| **mT0-3.7B** | mT5-xl | Encoder-decoder | 3.7B | 15B | mC4 | 6.4T | 1T | 1024 |
| **Instruct-XMLR$_{XL}$** | XML-R$_{XL}$ | Encoder-only | 3.5B | 0.6B | CC100 | 167B | 0.5T | 512 |

Table 1: Comparison between our model Instruct-XMLR$_{XL}$ and other baseline models. **Tok.** denotes the training tokens (billion). **Len.** denotes the sequence length limitation.

**Tasks and Datasets.** Following the previous work (Muennighoff et al., 2022), we evaluate the model's ability of task generalization in three held-out tasks that are not contained in the fine-tuned instruction data: conference resolution, sentence completion, and natural language inference (NLI). Specifically, we use XWinograd (Tikhonov & Ryabinin, 2021), XCOPA (Ponti et al., 2020), XNLI (Conneau et al., 2018) dataset for each task, respectively. Besides, we also evaluate in machine translation as a language generation task, which can also serve as a held-in task since xP3 contains various translation samples in multiple languages sampled from Flores-200 datasets (Costa-jussà et al., 2022). We use WMT'14 translation datasets[2] which contains five language splits between English and others. We adopt the test set for evaluation, each containing around 3,000 sentence pairs. Fortunately, all these datasets contain the language splits that do not exist in xP3, so we can also study and evaluate the model's ability for language generalization.

**Evaluation Settings.** During evaluation, we select five prompts from PromptSource (Bach et al., 2022) and then use them for all language splits of each dataset mentioned above. Finally, we report the average performance of these prompts. Since our model need to get target length in advance, we adopt length prediction module for WMT'14 datasets and fixed length for others. Besides, we should adopt some constraints on the design of prompts depending on the specific task formats due to the length prediction, more information can be found in Section 5.1.

## 5   ON THE POTENTIAL OF BERT FAMILY FOR INSTRUCTION FOLLOWING

In this section, we first present the potential of length prediction in successfully applying BERT family for instruction following. We then present the overall performance on three held-out tasks under the zero-shot setting to evaluate the model's ability task generalization. Finally, we further study the scaling effects, which serves as an important aspect in assessing the potential for further improvement of model capabilities and performance.

### 5.1   LENGTH PREDICTION

For several non-autoregressive models, length prediction is an extra task to determine the target sequence length during inference (Ghazvininejad et al., 2019; Qian et al., 2021). In comparison, the autoregressive models, e.g., the two baseline models BLOOMZ and mT0, generate the texts one-by-one in a left-to-right manner, and they can dynamically finish generation when meeting a special token indicating the end of sentences (e.g., [EOS]). For Instruct-XMLR$_{XL}$ which adopts Mask-Predict algorithm (Ghazvininejad et al., 2019) for generating the texts as mentioned in Section 3.3, length prediction is an essential process and directly related to the quality of final generation results. Next, we discuss the potential of length prediction through our experiments.

Adopting the length predicted by the model-self is a common practice in original task-specific non-autoregressive models for various generation tasks, e.g., machine translation, summarization, and story generation. In their experiments, length prediction only brings a tiny decline in performance compared with directly using the target length. It is also explainable that since these generation tasks are flexible, word adjustments can eliminate the effect of minor length differences during generation. Unlike several above-mentioned generation tasks with flexible target lengths, some tasks with determined target lengths may heavily rely on the length prediction. To verify this, we choose the XWinograd task and the prompt as {`The _ in the sentence below refers to {option1}. True or False? {sentence}` }, the label space is {`True, False`}. Notice these two labels have different lengths after the `sentencepiece`

---

[2]https://www.statmt.org/wmt14/translation-task.html

tokenizer in Instruct-XMLR$_{XL}$. As a result, once we set the length as one in advance during inference, the predictions are always {True}. If we set the length as two, the predictions are always {False}, indicating that the predicted length directly determines the label. This is not a reasonable situation and should be avoided in practical applications. As a result, we can add some constraints when selecting the prompts for these tasks.

Specifically, we can adopt length prediction module to get target length for traditional language generation tasks (e.g., machine translation since their target length can be flexible) and adopt a fixed length for determined tasks (e.g., multiple choice tasks since they always have the target length as one). However, for such tasks whose labels have determined but different lengths, such as the task with label space {False, True}, we can transform the label space into {Yes, No} whose labels have the same length, and then adopt the corresponding fixed length without leaking information about target labels.

## 5.2 OVERALL PERFORMANCE

Many works have shown that large language models after multitask instruction tuning can solve completely new tasks (Wei et al., 2021; Taori et al., 2023; Wang et al., 2022), we also examine our fine-tuned model: if Instruct-XMLR can successfully understand and complete the tasks which are not included in

| Models | XCOPA | XNLI | XWinograd |
|---|---|---|---|
| BLOOMZ-3B | 0.519 | 0.482 | 0.523 |
| mT0-3.7B | 0.627 | **0.528** | **0.584** |
| Instruct-XMLR$_{XL}$ | **0.646** | 0.518 | **0.582** |

Table 2: Accuracy on three held-out tasks.

the fine-tuning process. Table 2 presents the results. To avoid the influence of language generalization discussed in Section 6.1, we only choose the languages included in the fine-tuned data here. We report the average accuracy of 5 prompts in all tasks. We can find that Instruct-XMLR also demonstrates strong task generalization ability. After fine-tuning only 1/25 tokens of baseline models, Instruct-XMLR can significantly outperform a decoder-only model with comparable size BLOOMZ-3B in all tasks. Compared with the more competitive model mT0-3.7B, Instruct-XMLR achieves better performance on XCOPA and comparable performance on XWinograd, but underperforms a little on XNLI. We attribute this failure to: (1) XNLI is a multilingual dataset for traditional natural language inference (NLI) task, and mT0 with the encoder-decoder architecture is more beneficial to this task as also mentioned in Muennighoff et al. (2022), and (2) mT0-3.7B is trained more much longer in both pre-training (1 trillion v.s. 0.5 trillion tokens) and instruction tuning stage (15 billion v.s. 0.6 billion tokens), which can boost the performance of NLI tasks (Goyal et al., 2021; Hoffmann et al., 2022).

## 5.3 SCALING EFFECTS

Scaling law plays a vital role in the recent success of LLMs (Hoffmann et al., 2022; Touvron et al., 2023). Since XML-R has different versions containing various parameters, we study the performance changes as the model size increases. Besides, we also focus on another layer of scaling, the number of training tokens during the fine-tuning process. Firstly, we also conduct experiments on relatively small models, XML-R$_{Base}$ and XML-R$_{Large}$ with 270M and 550M parameters, respectively. Table 3 presents the results. We can find an apparent growing trend as the model parameters increase. After fine-tuned with the same data, Instruct-XMLR$_{XL}$ outperforms Instruct-XMLR$_{Base}$ and Instruct-XMLR$_{Large}$ on all tasks, indicating the important role of model size in task generalization. Then, we focus on the scaling effects of training tokens during instruction tuning process. Figure 2 plots the performance changes in the training process. The performance of all tasks keeps improving as the training continues, which is consistent with our intuition. Overall, Instruct-XMLR also demonstrates the positive scaling effects, which presents potential improvements in the future.

## 6 POSSIBLE LIMITATIONS, REASONS AND SOLUTIONS

In Section 5, we have noticed the potential of the BERT family for instruction following. However, more aspects remain to be explored for Instruct-XMLR before becoming a superior instruction follower. In this section, we investigate some limitations that may be caused by the specific factors of

| Models | XCOPA | XNLI | XWinograd |
|---|---|---|---|
| Instruct-XMLR$_{\text{Base}}$ | 0.583 | 0.386 | 0.521 |
| Instruct-XMLR$_{\text{Large}}$ | 0.602 | 0.432 | 0.545 |
| Instruct-XMLR$_{\text{XL}}$ | **0.646** | **0.518** | **0.582** |

Table 3: Effects on models scaling.

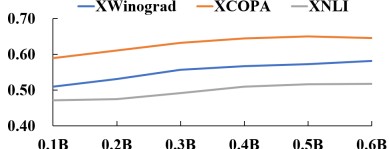

Figure 2: Effects on data scaling.

our model, e.g., the backbone model, including the original architecture and training corpus, the traditional non-autoregressive generation manner we adopted during inference, etc. Then, after finding these limitations, we also give some possible solutions.

## 6.1 LANGUAGE GENERALIZATION

Since Instruct-XMLR and two baseline models are multilingual, they should support multiple languages. Naturally, we aim to explore their abilities of language generalization, i.e., whether the model can perform well in languages not contained in fine-tuned datasets. Language generalization is also a widespread concern of language models and is related strongly to language distribution in the training data.

We select the language splits that are not included in the fine-tuned data of each task for evaluation. Table 4 presents the results. We can find that Instruct-XMLR outperforms BLOOMZ-3B but fails behind mT0-3.7B in all tasks, especially on XNLI dataset. Considering the same language composition in fine-tuning datasets, we explain this to the difference between their pre-training stages. Firstly, all these

| Models | XCOPA | XNLI | XWinograd |
|---|---|---|---|
| BLOOMZ-3B | 0.427 | 0.391 | 0.499 |
| mT0-3.7B | **0.605** | **0.533** | **0.569** |
| Instruct-XMLR$_{\text{XL}}$ | 0.599 | 0.466 | 0.539 |

Table 4: Results of language generalization.

languages are outside the pre-training data of BLOOMZ-3B, the declines for BLOOMZ-3B are easy to understand. Regarding Instruct-XMLR and mT0-3.7B, these languages are all in the pre-training data and show a similar composition distribution. As shown in Table 1, mT0-3.7B is trained with twice as many tokens than Instruct-XMLR and learns more robust language knowledge. As a result, Instruct-XMLR is inferior to mT0-3.7B in the ability of language generalization.

## 6.2 GENERATION TASKS

As mentioned in numerous previous works (Gu et al., 2018; Xiao et al., 2022), the non-autoregressive models remove the target-side dependency during training. They will result in a performance drop in generation quality compared with the traditional autoregressive models. As Instruct-XMLR also adopts

| Models | EN → X | X → EN | Speedup |
|---|---|---|---|
| BLOOMZ-3B | 4.15 | 8.30 | 1.1x |
| mT0-3.7B | **9.41** | **22.82** | 1.0x |
| Instruct-XMLR$_{\text{XL}}$ | 5.72 | 13.51 | **3.1x** |
| Instruct-XMLR$_{\text{XL}}$-MT | 8.83 | 18.31 | 3.0x |

Table 5: Results of machine translation tasks.

the non-autoregressive manner to generate texts, this performance decline still exists. What's more, unlike the generative pre-training of baseline models, the pre-training paradigm of the BERT family is not designed for language generation, leading to difficulty in learning well just in the instruction tuning process. To evaluate the generation ability of Instruct-XMLR, we choose WMT'14 translation datasets, which are widely used in non-autoregressive works. Table 5 presents the results. We use SacreBLEU [3] (Post, 2018) as our evaluation metric for all translation directions. We split the dataset into two subsets whose source and target language is English (EN→X and X→EN). We report the average performance of 5 prompts. Besides we also report the speedup by computing the number of tokens per second when translating the texts. As mT0-3.7B is the lowest, the corresponding speedup is 1.0x. Results show that although Instruct-XMLR$_{\text{XL}}$ can speed up the decoding by around 3 times that baselines due to the non-autoregressive manner, the performance drops seriously compared with mT0-3.7B.

---

[3] https://github.com/mjpost/sacrebleu

We further analyze the main internal reasons: non-autoregressive generation manner or the pre-training paradigm. As mentioned in (Muennighoff et al., 2022), our fine-tuned dataset contains much more short target texts for training. Thus, Instruct-XMLR$_{XL}$ still can not learn superior generation abilities. As a result, we aim to enhance the learning on generation tasks during the instruction tuning process. Specifically, we random sample a subset containing all translation pairs but keep the language distribution the same as the original one. We get Instruct-XMLR$_{XL}$-MT after fine-tuning on this purer instruction dataset with only a quarter updates steps of Instruct-XMLR$_{XL}$. We can witness a significant performance improvement, indicating that the pre-training paradigm is more decisive. Therefore, it is necessary to enhance the learning on generation tasks for the BERT family; even a suitable generative post-training process is optimal before instruction tuning.

### 6.3 UNRELIABLE PREDICTED LENGTH

As mentioned in section 5.1, we should transform the label space by designing the prompts for several tasks. However, we also recognize that this can not be adapted to all tasks in practical applications. For example, it is unrealizable for a language modeling task whose targets have different lengths of various samples. Besides, length prediction also seems not suitable since the length of each sample is determined. Therefore, we conduct quantitative experiments to analyze the effects of length prediction in various

| Settings | Accuracy | Gap |
|:--------:|:--------:|:----:|
| A | 0.87 | 0.02 |
| B | 0.42 | 33.2 |
| C | 0.05 | 24.3 |

Table 6: Effects on length.

tasks with determined target length. Specifically, we select the following evaluation settings: (1) multiple choice questions task with options {A, B, C, D}, denoted as setting A; (2) the specific XWinograd task mentioned above with label space as {True, False} without transforming the label space, denoted as setting B; (3) a language modeling task LAMBADA (Paperno et al., 2016), where the target words to be generated are with various length, this is a more challenging setting, denoted as setting C. Table 6 presents the results. We report the accuracy of length prediction (the column Accuracy) and the performance gap compared with using fixed target length or the ground length (the column Gap). We can find that: length prediction is unreliable when meeting challenging testing scenarios. For setting A, since the model may predict the incorrect length, e.g., one as two, the result cab be {C:} or {C.}, which is equal to the label {C}. For settings B and C, length prediction almost fails, leading to a seriously declining performance. Therefore, we look forward to a more robust mechanism to determine the target length for non-aoturegressive models.

### 6.4 FEW-SHOT PROMPTING LEARNING

Few-shot prompting, also known as in-context learning (ICL), is first mentioned in GPT-3 (Brown et al., 2020) and allows the LLMs to learn specific abilities to solve different tasks with only a few examples concatenated in the demonstration. Then, LLMs can perform well in various without updating model parameters. ICL has become a mainstream method to boost the performance of various LLMs. Unfortunately, Instruct-XMLR has not yet demonstrated this ability during our experiments. We also concatenate several examples simply before the test example. However, this setting confuses the model and results in disorganized predictions. We attribute this to the bi-directional nature of BERT family since ICL has only been proven effective in autoregressive unidirectional language model. Therefore, we need more efforts to explore this and leave it as a future work.

## 7 CONCLUSION

This paper thoroughly explores the potential and limitations of BERT family as instruction followers. We find that our fine-tuned Instruct-XMLR can also generalize well across tasks and languages compared with current popular LLMs of model architecture with comparable model parameters. Besides, we also witnessed several limitations through our experiments, such as the performance decline of long-generation task and failure in few-shot prompting learning. We further analyze the reasons and point out that some of the problems are possibly owing to the limited capabilities of the backbone models since there are no previous BERT family models that are as powerful as the decoder-only and encoder-decoder counterparts yet. Therefore, we hope that researchers can pay more attention to BERT family, and make them become competitive members in the family of current large language models.

## ACKNOWLEDGMENTS

We would like to thank the anonymous reviewers for their constructive comments. This work was supported by the National Science Foundation of China (NSFC No. 62206194), the Natural Science Foundation of Jiangsu Province, China (Grant No. BK20220488), and the Young Elite Scientists Sponsorship Program by CAST (Grant No. 2023QNRC001).

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

# A APPENDIX

## A.1 PROMPTS

We provide the prompts adopted in our main experiments here.

| Task | Index | Prompt Template |
|---|---|---|
| XNLI | ① | Suppose {{premise}} Can we infer that "{{hypothesis}}"? Yes, no, or maybe? |
| | ② | {{premise}} Are we justified in saying that "{{hypothesis}}"? Yes, no, or maybe? |
| | ③ | Given that premise Does it follow that "{{hypothesis}}" Yes, no, or maybe? |
| | ④ | Given that {{premise}} Therefore, it must be true that "{{hypothesis}}"? Yes, no, or maybe? |
| | ⑤ | {{premise}} Based on the previous passage, is it true that "{{hypothesis}}"? Yes, no, or maybe? |
| XCOPA | ① | {{premise}} Select the most plausible effect/cause: {{Option 1}} {{Option 2}} |
| | ② | {{premise}} As a consequence.../This happened because... Help me pick the more plausible option: {{Option 1}} {{Option 2}} |
| | ③ | {{premise}} I am hesitating between two options. Help me choose the best effect/cause: {{Option 1}} {{Option 2}} |
| | ④ | {{premise}} We are looking for an effect/cause, what is the best option? {{Option 1}} {{Option 2}} |
| | ⑤ | {{Option 1}} {{Option 2}} {{premise}} because/so: |
| XWinogra | ① | {{premise}} The _ in the sentence above refers to {{Option 1}} Yes or No? |
| | ② | {{premise}} What does the _ in the above sentence refer to? {{Option 1}} or {{Option 2}}? |
| | ③ | {{premise}} The _ in the sentence above refers to {{Option 1}} Yes or No? |
| | ④ | What does the _ stand for in the below sentence? {{premise}} {{Option 1}} or {{Option 2}} |
| | ⑤ | {{premise}} Replace the _ in the above sentence with the correct option: {{Option 1}} or {{Option 2}} |
| WMT'14 Translation | ① | A text in {{src-lang}}: {{Source}} The same text in {{tgt-lang}} |
| | ② | Translate the following text from {{src-lang}} to {{tgt-lang}}: {{Source}} |
| | ③ | {{Source}} Here is the same text in {{tgt-lang}}: |
| | ④ | {{Source}} The previous text is in {{src-lang}} Here is the same text in {{tgt-lang}}: |
| | ⑤ | Text in {{src-lang}}: {{Source}} Translation of the previous text to {{tgt-lang}}: |

Table 7: Different prompts adopted in our experiments. Texts within quotes denote the input sequences. {{src-lang}} and {{tgt-lang}} denote the source and target languages.

## A.2 RESULTS

As we only report the average performance in the main body of our paper, we present different results based on the different prompts and different languages splits here.

| Models | Prompts | EN | FR | PT | ZH | JP* | RU* |
|---|---|---|---|---|---|---|---|
| BLOOMZ-3B | ① | 0.511 | 0.530 | 0.475 | 0.474 | 0.452 | 0.454 |
| | ② | 0.548 | 0.482 | 0.525 | 0.536 | 0.490 | 0.524 |
| | ③ | 0.557 | 0.530 | 0.502 | 0.567 | 0.485 | 0.498 |
| | ④ | 0.501 | 0.494 | 0.483 | 0.524 | 0.503 | 0.540 |
| | ⑤ | 0.592 | 0.494 | 0.529 | 0.609 | 0.510 | 0.533 |
| mT0-3B | ① | 0.568 | 0.651 | 0.551 | 0.583 | 0.560 | 0.527 |
| | ② | 0.575 | 0.566 | 0.574 | 0.575 | 0.569 | 0.571 |
| | ③ | 0.619 | 0.566 | 0.597 | 0.585 | 0.603 | 0.590 |
| | ④ | 0.579 | 0.554 | 0.582 | 0.502 | 0.560 | 0.546 |
| | ⑤ | 0.615 | 0.590 | 0.608 | 0.629 | 0.597 | 0.568 |
| Instruct-XMLR$_{XL}$ | ① | 0.553 | 0.518 | 0.532 | 0.554 | 0.550 | 0.470 |
| | ② | 0.563 | 0.619 | 0.570 | 0.607 | 0.539 | 0.505 |
| | ③ | 0.567 | 0.566 | 0.548 | 0.627 | 0.565 | 0.517 |
| | ④ | 0.568 | 0.602 | 0.567 | 0.647 | 0.592 | 0.502 |
| | ⑤ | 0.637 | 0.578 | 0.589 | 0.631 | 0.600 | 0.546 |

Table 8: Results of conference resolution task on XWinograd datasets. The language with * is not contained in fine-tuned dataset.

| Models | Prompts | EN→X | | | | | X→EN | | | | |
|---|---|---|---|---|---|---|---|---|---|---|---|
| | | FR | HI | DE | CS | RU | FR | HI | DE | CS | RU |
| BLOOMZ-3B | ① | 10.12 | 2.61 | 0.70 | 0.19 | 0.28 | 8.15 | 5.46 | 4.91 | 2.15 | 3.77 |
| | ② | 32.65 | 6.57 | 2.64 | 0.16 | 0.48 | 33.13 | 17.55 | 15.18 | 2.57 | 13.97 |
| | ③ | 7.99 | 1.93 | 0.53 | 0.23 | 0.29 | 8.17 | 5.44 | 4.59 | 2.24 | 3.55 |
| | ④ | 7.07 | 1.79 | 0.43 | 0.24 | 0.25 | 6.76 | 4.67 | 4.12 | 1.95 | 3.19 |
| | ⑤ | 20.98 | 3.47 | 1.56 | 0.14 | 0.54 | 24.32 | 12.56 | 8.92 | 1.59 | 8.57 |
| mT0-3B | ① | 24.9 | 8.41 | 6.62 | 5.26 | 7.71 | 28.78 | 19.63 | 24.35 | 21.36 | 23.82 |
| | ② | 27.33 | 9.58 | 7.01 | 4.26 | 9.13 | 31.52 | 20.14 | 27.28 | 28.05 | 27.85 |
| | ③ | 24.53 | 7.93 | 6.72 | 4.64 | 7.39 | 28.35 | 19.56 | 23.98 | 24.37 | 23.41 |
| | ④ | 22.12 | 8.32 | 7.01 | 5.03 | 7.76 | 28.34 | 19.81 | 24.00 | 24.37 | 23.42 |
| | ⑤ | 11.78 | 3.73 | 5.65 | 0.60 | 1.77 | 19.55 | 10.36 | 17.76 | 15.91 | 14.63 |
| Instruct-XMLR$_{XL}$ | ① | 14.97 | 7.03 | 5.23 | 2.57 | 0.75 | 19.59 | 13.16 | 15.83 | 14.94 | 7.62 |
| | ② | 12.26 | 4.83 | 2.67 | 2.18 | 0.73 | 18.57 | 12.18 | 14.62 | 14.05 | 1.80 |
| | ③ | 15.84 | 7.00 | 5.20 | 2.49 | 0.73 | 20.53 | 13.72 | 16.52 | 15.38 | 8.27 |
| | ④ | 14.99 | 7.16 | 5.42 | 2.69 | 0.77 | 17.79 | 12.29 | 13.81 | 13.34 | 5.89 |
| | ⑤ | 12.73 | 6.51 | 4.99 | 2.53 | 0.76 | 19.33 | 12.84 | 14.92 | 14.03 | 6.79 |
| Instruct-XMLR$_{XL}$-MT | ① | 16.53 | 7.75 | 8.02 | 5.77 | 4.25 | 21.88 | 13.35 | 17.83 | 16.92 | 16.32 |
| | ② | 20.76 | 9.52 | 9.95 | 7.20 | 5.83 | 26.35 | 19.48 | 23.71 | 25.00 | 22.12 |
| | ③ | 16.23 | 7.94 | 8.35 | 5.74 | 4.14 | 19.78 | 11.26 | 16.23 | 17.31 | 15.32 |
| | ④ | 16.42 | 7.74 | 8.16 | 5.47 | 3.51 | 19.32 | 13.98 | 17.69 | 16.53 | 15.34 |
| | ⑤ | 16.60 | 7.82 | 7.59 | 5.44 | 4.06 | 22.13 | 15.33 | 18.64 | 17.36 | 18.65 |

Table 9: Results of machine translation task on WMT'14 datasets.

| Models | Prompts | AR | EN | ES | FR | HI | SW | UR | VI | ZH | BG* | DE* | EL* | RU* | TH* | TR* |
|---|---|---|---|---|---|---|---|---|---|---|---|---|---|---|---|---|
| BLOOMZ-3B | ① | 0.482 | 0.540 | 0.506 | 0.514 | 0.468 | 0.418 | 0.437 | 0.462 | 0.477 | 0.394 | 0.408 | 0.404 | 0.437 | 0.377 | 0.353 |
| | ② | 0.483 | 0.522 | 0.503 | 0.505 | 0.462 | 0.419 | 0.429 | 0.466 | 0.490 | 0.411 | 0.410 | 0.408 | 0.439 | 0.429 | 0.363 |
| | ③ | 0.488 | 0.518 | 0.505 | 0.516 | 0.463 | 0.417 | 0.428 | 0.485 | 0.478 | 0.379 | 0.402 | 0.367 | 0.418 | 0.361 | 0.348 |
| | ④ | 0.490 | 0.504 | 0.498 | 0.510 | 0.455 | 0.419 | 0.434 | 0.457 | 0.473 | 0.380 | 0.398 | 0.381 | 0.412 | 0.379 | 0.356 |
| | ⑤ | 0.518 | 0.570 | 0.535 | 0.536 | 0.495 | 0.433 | 0.458 | 0.516 | 0.508 | 0.380 | 0.422 | 0.365 | 0.421 | 0.369 | 0.356 |
| mT0-3B | ① | 0.518 | 0.568 | 0.542 | 0.538 | 0.498 | 0.492 | 0.479 | 0.531 | 0.506 | 0.533 | 0.533 | 0.533 | 0.505 | 0.521 | 0.487 |
| | ② | 0.509 | 0.552 | 0.548 | 0.542 | 0.502 | 0.496 | 0.467 | 0.532 | 0.506 | 0.542 | 0.548 | 0.538 | 0.531 | 0.510 | 0.506 |
| | ③ | 0.539 | 0.568 | 0.539 | 0.562 | 0.512 | 0.520 | 0.490 | 0.549 | 0.547 | 0.550 | 0.535 | 0.551 | 0.539 | 0.548 | 0.535 |
| | ④ | 0.514 | 0.578 | 0.539 | 0.554 | 0.489 | 0.518 | 0.458 | 0.524 | 0.529 | 0.528 | 0.534 | 0.539 | 0.508 | 0.516 | 0.513 |
| | ⑤ | 0.538 | 0.583 | 0.556 | 0.558 | 0.533 | 0.526 | 0.514 | 0.547 | 0.544 | 0.560 | 0.565 | 0.558 | 0.549 | 0.553 | 0.537 |
| Instruct-XMLR$_{XL}$ | ① | 0.515 | 0.566 | 0.547 | 0.541 | 0.504 | 0.494 | 0.488 | 0.530 | 0.513 | 0.457 | 0.504 | 0.443 | 0.453 | 0.407 | 0.500 |
| | ② | 0.499 | 0.559 | 0.545 | 0.538 | 0.510 | 0.496 | 0.492 | 0.529 | 0.518 | 0.459 | 0.511 | 0.459 | 0.459 | 0.419 | 0.508 |
| | ③ | 0.514 | 0.555 | 0.535 | 0.540 | 0.503 | 0.492 | 0.478 | 0.523 | 0.518 | 0.453 | 0.507 | 0.440 | 0.442 | 0.406 | 0.506 |
| | ④ | 0.506 | 0.554 | 0.539 | 0.529 | 0.504 | 0.487 | 0.480 | 0.521 | 0.515 | 0.455 | 0.508 | 0.456 | 0.461 | 0.417 | 0.497 |
| | ⑤ | 0.502 | 0.558 | 0.532 | 0.536 | 0.507 | 0.485 | 0.486 | 0.522 | 0.504 | 0.473 | 0.505 | 0.458 | 0.477 | 0.426 | 0.500 |

Table 10: Results of natural language inference task on XNLI datasets.

| Models | Prompts | ID | SE | TA | VI | ZH | ET* | HT* | IT* | QU* | TH* | TR* |
|---|---|---|---|---|---|---|---|---|---|---|---|---|
| BLOOMZ-3B | ① | 0.74 | 0.51 | 0.57 | 0.76 | 0.76 | 0.43 | 0.53 | 0.50 | 0.47 | 0.46 | 0.48 |
| | ② | 0.62 | 0.44 | 0.53 | 0.68 | 0.67 | 0.46 | 0.44 | 0.49 | 0.43 | 0.46 | 0.50 |
| | ③ | 0.65 | 0.47 | 0.52 | 0.70 | 0.72 | 0.51 | 0.49 | 0.53 | 0.38 | 0.48 | 0.55 |
| | ④ | 0.50 | 0.42 | 0.43 | 0.54 | 0.54 | 0.43 | 0.42 | 0.45 | 0.42 | 0.45 | 0.47 |
| | ⑤ | 0.14 | 0.25 | 0.36 | 0.13 | 0.32 | 0.37 | 0.32 | 0.07 | 0.38 | 0.18 | 0.25 |
| mT0-3B | ① | 0.76 | 0.61 | 0.66 | 0.69 | 0.74 | 0.69 | 0.68 | 0.70 | 0.49 | 0.69 | 0.71 |
| | ② | 0.77 | 0.57 | 0.70 | 0.67 | 0.79 | 0.68 | 0.70 | 0.70 | 0.50 | 0.71 | 0.69 |
| | ③ | 0.72 | 0.58 | 0.69 | 0.67 | 0.76 | 0.72 | 0.67 | 0.68 | 0.45 | 0.73 | 0.68 |
| | ④ | 0.75 | 0.59 | 0.66 | 0.68 | 0.74 | 0.70 | 0.65 | 0.67 | 0.48 | 0.69 | 0.70 |
| | ⑤ | 0.40 | 0.36 | 0.39 | 0.39 | 0.33 | 0.44 | 0.45 | 0.35 | 0.34 | 0.42 | 0.38 |
| Instruct-XMLR$_{XL}$ | ① | 0.82 | 0.58 | 0.71 | 0.75 | 0.74 | 0.68 | 0.57 | 0.73 | 0.58 | 0.72 | 0.68 |
| | ② | 0.73 | 0.60 | 0.71 | 0.75 | 0.70 | 0.71 | 0.57 | 0.70 | 0.63 | 0.71 | 0.64 |
| | ③ | 0.66 | 0.61 | 0.67 | 0.71 | 0.69 | 0.62 | 0.54 | 0.68 | 0.58 | 0.66 | 0.68 |
| | ④ | 0.77 | 0.62 | 0.67 | 0.76 | 0.68 | 0.70 | 0.54 | 0.65 | 0.63 | 0.65 | 0.64 |
| | ⑤ | 0.43 | 0.53 | 0.40 | 0.40 | 0.47 | 0.43 | 0.47 | 0.37 | 0.41 | 0.42 | 0.39 |

Table 11: Results of sentence completion task on XCOPA datasets.

## A.3 The Performance Variance of Different Prompts.

Since we adopt five different prompts for each task, analyzing how stable these models perform across these different prompts is essential. As a result, we present the performance variance of difference prompts here. Table 12, Table 13 and Table 14 present the results. We report the performance range, i.e., the performance boundary, to show the performance variance of difference prompts. We can find that: (1) The performance difference across prompts of Instruct-XMLR is nearly compared with two baseline models in all tasks and even smaller in XCOPA, XNLI, and machine translation tasks, indicating that our method does not bring instability across prompts. (2) The performance difference in XCOPA is relatively big, but this is the case for all models. Instruct-XMLR performs better than other models.

| Models | XCOPA | XNLI | XWinograd |
|---|---|---|---|
| BLOOMZ-3B | 0.519 $\pm_{0.289}$ | 0.482 $\pm_{0.026}$ | 0.523 $\pm_{0.033}$ |
| mT0-3.7B | 0.627 $\pm_{0.253}$ | **0.528** $\pm_{0.016}$ | **0.584** $\pm_{0.030}$ |
| Instruct-XMLR$_{XL}$ | **0.646** $\pm_{0.200}$ | 0.518 $\pm_{0.004}$ | **0.582** $\pm_{0.043}$ |

Table 12: Accuracy on three held-out tasks.

| Models | XCOPA | XNLI | XWinograd |
|---|---|---|---|
| BLOOMZ-3B | 0.427 $\pm_{0.165}$ | 0.391 $\pm_{0.019}$ | 0.499 $\pm_{0.046}$ |
| mT0-3.7B | **0.605** $\pm_{0.208}$ | **0.533** $\pm_{0.021}$ | **0.569** $\pm_{0.028}$ |
| Instruct-XMLR$_{XL}$ | 0.599 $\pm_{0.184}$ | 0.466 $\pm_{0.007}$ | 0.539 $\pm_{0.034}$ |

Table 13: Results of language generalization.

| Models | EN $\rightarrow$ X | X $\rightarrow$ EN |
|---|---|---|
| BLOOMZ-3B | 4.15 $\pm_{4.35}$ | 8.30 $\pm_{8.18}$ |
| mT0-3.7B | **9.41** $\pm_{4.70}$ | **22.82** $\pm_{7.18}$ |
| Instruct-XMLR$_{XL}$ | 5.72 $\pm_{1.19}$ | 13.51$\pm_{1.37}$ |
| Instruct-XMLR$_{XL}$-MT | 8.83 $\pm_{1.82}$ | 18.31 $\pm_{5.02}$ |

Table 14: Results of machine translation tasks.

## A.4 Evaluation on More Tasks and Datasets.

Beyond the tasks shown in 4.2, we also evaluate more tasks and datasets. Table 15 and Table 16 present the corresponding results.

**Tasks and Datasets.** We evaluate our models in more tasks, including language generation and understanding. For language generation, except the machine translation task we have included in Table 5, we select various tasks with different characteristics, e.g., text summarization task which has relatively long sequences, text simplification and paraphrase generation tasks whose inputs and targets have the similar expression, controllable generation task which requires the model to generates texts meeting certain controllable constraints as humans wish reliably, and dialogue generation task which has relatively flexible outputs. For text summarization task, we use XSUM (Narayan et al., 2018) and Gigaword (Rush et al., 2015), which contain the online articles and single sentence summary pairs from the British Broadcasting Corporation and summary from Gigaword corpus, respectively. For text simplification task, we adopt WIKI-AUTO (Jiang et al., 2020) which contains aligned sentences from English Wikipedia and Simple English Wikipedia. We adopt Quora Question Pairs (QQP) for paraphrase generation task, where each question pair is semantically equivalent. For controllable generation task, we adopt COMMONGEN (Lin et al., 2020) in which the model needs to generate a coherent sentence using the given common concepts. For dialogue generation task, we

| | Text Summarization | | | |
| | XSUM | | Gigaword | |
| | Rouge-1 | Rouge-L | Rouge-1 | Rouge-L |
|---|---|---|---|---|
| Bloomz-3B | 35.12 ±11.32 | 27.97 ±9.35 | 31.45 ±10.60 | 28.94 ±10.04 |
| mT0-3.7B | **37.85** ±0.04 | **30.28** ±0.26 | **34.31** ±0.18 | **31.54** ±0.10 |
| Instruct-XMLR$_{XL}$ | 35.62 ±0.22 | 27.90 ±0.23 | 30.64 ±0.11 | 28.52 ±0.21 |
| Instruct-XMLR$_{XL}$ w/GL | 36.00 ±0.19 | 27.84 ±0.10 | 32.60 ±0.55 | 29.85 ±0.60 |
| | Text Simplification | | Paraphrase Generation | |
| | WIKI-AUTO | | QQP | |
| | Bleu-2 | Distinct-2 | Bleu-2 | Distinct-2 |
| Bloomz-3B | 41.90 ±6.02 | 0.57 ±0.03 | 38.24 ±10.79 | 0.54 ±0.28 |
| mT0-3.7B | 55.92 ±1.83 | 0.64 ±0.01 | 51.93 ±10.39 | 0.63 ±0.08 |
| Instruct-XMLR$_{XL}$ | 47.91 ±1.85 | 0.65 ±0.01 | 43.50 ±2.49 | 0.67 ±0.01 |
| Instruct-XMLR$_{XL}$ w/GL | **56.93** ±2.00 | **0.76** ±0.00 | **60.02** ±0.72 | **0.69** ±0.01 |
| | Controllable Generation | | Dialogue Generation | |
| | COMMONGEN | | PersonaChat | |
| | Bleu-2 | Distinct-2 | Bleu-2 | Distinct-2 |
| Bloomz-3B | 35.20 ±0.26 | 0.44 ±0.01 | 16.39 ±0.04 | 0.39 ±0.01 |
| mT0-3.7B | 31.86 ±0.14 | 0.36 ±0.04 | 9.17 ±0.02 | 0.39 ±0.01 |
| Instruct-XMLR$_{XL}$ | 21.68 ±0.85 | **0.52** ±0.02 | 2.88 ±0.04 | 0.52 ±0.03 |
| Instruct-XMLR$_{XL}$ w/GL | **35.44** ±1.03 | 0.48 ±0.01 | **17.30** ±0.01 | **0.53** ±0.01 |

Table 15: Results on various language generation tasks.

| | Fact Verification | Sentiment Analysis | |
| | FEVER | SST-2 | Poem |
| | Accuracy (%) | Accuracy (%) | Accuracy (%) |
|---|---|---|---|
| Bloomz-3B | 81.23 ±4.67 | **95.40** ±0.30 | 18.67 ±2.33 |
| mT0-3.7B | 83.47 ±3.37 | 82.30 ±1.50 | 14.67 ±1.67 |
| Instruct-XMLR$_{XL}$ | **88.10** ±1.20 | 92.20 ±0.60 | **25.33** ±0.67 |
| Instruct-XMLR$_{XL}$ w/GL | **88.10** ±1.20 | 92.20 ±0.60 | **25.33** ±0.67 |

Table 16: Results on various language understanding tasks.

adopt PersonaChat (Zhang et al., 2018), which contains around 150k data triples formatted as (profile, conversation, response). For language understanding tasks, we select FEVER (Thorne et al., 2018) for the fact verification task, which aims to verify the correctness of textual claims against textual sources. We also use SST-2 (Socher et al., 2013) and Poem (Sheng & Uthus, 2020) datasets to evaluate the sentiment analysis task, which analyzes the sentiment information given the specific sentence.

**Evaluation Metrics.** We utilize various evaluation metrics for these tasks. For text summarization task, we adopt ROUGE F1 score (Lin & Hovy, 2002) following previous works. For other language generation tasks, i.e., text simplification, paraphrase generation, controllable generation, and dialogue generation, we adopt BLEU (Papineni et al., 2002) and Distinct (Li et al., 2015) to measure the n-gram level precision and the diversity of the generated texts. For language understanding tasks, we report the corresponding accuracy.

**Results.** As shown in Table 15 and Table 16, except the performance of standard Instruct-XMLR, we also report the performance of Instruct-XMLR w/GL which decodes that we use ground truth length rather than the predicted length during inference. We aim to analyze the effects of length prediction for different tasks. We can mainly find that: (1) Instruct-XMLR is more suitable for language understanding tasks. It outperforms mT0 and BLOOMZ in most testing scenarios (except underperforming BLOOMZ on sst-2). (2) We can notice two groups for language generation tasks according to the effects of length prediction. The length prediction is usable for text summarization, text simplification, and paraphrase generation tasks. The performance in these tasks of Instruct-XMLR outperforms BLOOMZ in most testing scenarios but underperforms mT0. This is the same case for the machine translation task as shown in Table 4 in the main body. (3) For other language generation tasks, i.e., COMMONGEN and PersonaChat, although Instruct-XMLR can achieve promising performance with ground truth length, adopting length prediction results in significant performance declines. This is easy to understand, i.e., these tasks have no strong alignment relationship between source length and target length. Hence, Instruct-XMLR fails to predict the reliable length based on the source representation. More reliable alternative solutions should be explored to determine the target length for these tasks in the future. (4) The Distinct score of Instruct-XMLR is superior in all generation tasks, indicating that Instruct-XMLR can generate more diverse texts for all text generation tasks. We attribute this to the non-autoregressive generation (Mask-Predict algorithm) generation manner where the dependency of different tokens is released compared with adopting the standard next token prediction algorithm in traditional autoregressive models.

## A.5 More Exploration for Few-shot Prompting Learning

As mentioned in Section 6.4, Instruct-XMLR has not yet demonstrated the ability of few-shot prompting during our experiments. Since this capacity is first noticed in GPT-3 (Brown et al., 2020), current researchers believe that few-shot prompting learning arises in the pre-training stage and benefits from its generative pre-training paradigm. However, the different pre-training paradigm of Instruct-XMLR (i.e., simply predicting the masked tokens without generative paradigm) fails to equip the capacity of few-shot prompting learning. As a result, We have tried to make up for this flaw by including some few-shot instruction data during training rather than the only zero-shot instruction data we adopted in our main experiments. Unfortunately, this also doesn't bring any profit. Table 17 and Table 18 present the corresponding results. We can find that adopting few-shot prompting learning even harms the overall performance.

| Models | XCOPA | XNLI | XWinograd |
|---|---|---|---|
| Instruct-XMLR$_{XL}$ w/ zs | 0.646 | 0.518 | 0.582 |
| w/ ICL 4 | 0.548 | 0.414 | 0.547 |
| Instruct-XMLR$_{XL}$ w/ fs | 0.642 | 0.489 | 0.565 |
| w/ ICL 4 | 0.571 | 0.442 | 0.553 |

Table 17: Accuracy on three held-out tasks. w/ zs denotes the model trained on zero-shot instruction data, w/ fs denotes the model trained on mixed zero/few-shot instruction data, w/ ICL 4 denotes that we adopt 4 examples in the demonstration for few-shot prompting learning.

| Models | XCOPA | XNLI | XWinograd |
|---|---|---|---|
| Instruct-XMLR$_{XL}$ w/ zs | 0.599 | 0.466 | 0.539 |
| w/ ICL 4 | 0.521 | 0.397 | 0.506 |
| Instruct-XMLR$_{XL}$ w/ fs | 0.591 | 0.451 | 0.528 |
| w/ ICL 4 | 0.536 | 0.419 | 0.514 |

Table 18: Results of language generalization. w/ zs denotes the model trained on zero-shot instruction data, w/ fs denotes the model trained on mixed zero/few-shot instruction data, w/ ICL 4 denotes that we adopt 4 examples in the demonstration for few-shot prompting learning.

## A.6   RESULTS OF XML-R WITHOUT INSTRUCTION TUNING

To analyze the effects of instruction tuning for our model, we compare Instruct-XMLR and the original pre-trained XML-R without instruction tuning. However, it is difficult or even impossible to collect the evaluation results of XMLR-w/o instruction tuning owing to its poor performance on these tasks. We provide a few representative examples for intuitive understanding in Table 19. We can find that XML-R fails to complete the task without instruction tuning, i.e., it always generates the special tokens `[PAD]` without actual meaning in XCOPA, XNLI, and XWinograd tasks and generates texts unrelated to the specific prompt in machine translation.

| Task | Type | Output |
|---|---|---|
| XCOPA | Source | Mata saya menjadi merah dan bengkak. Select the most plausible cause: - A: Saya tertangis. - B: Saya tertawa. |
|  | Target | A |
|  | Instruct-XMLR$_{XL}$ | A |
|  | XML-R$_{XL}$ | `[PAD]` |
| XNLI | Source | Suppose "And he said, Mama, I'm home." Can we infer that "He didn't say a word."? |
|  | Target | No |
|  | Instruct-XMLR$_{XL}$ | No |
|  | XML-R$_{XL}$ | `[PAD]` |
| XWinograd | Source | Joan made sure to thank Susan for all the help _ had given. The _ in the sentence above refers to Susan. Yes or No? |
|  | Target | Yes |
|  | Instruct-XMLR$_{XL}$ | Yes |
|  | XML-R$_{XL}$ | `[PAD]` |
| MT | Source | A text in German: Kinder brauchen Stabilität und Sicherheit. The same text in English: |
|  | Target | Children need stability and certainty. |
|  | Instruct-XMLR$_{XL}$ | Children need safety and security. |
|  | XML-R$_{XL}$ | Fragen und Antworten zum. |
| MT | Source | Translate the following text from German to English: Auch Fast Food ist erlaubt. |
|  | Target | Fast food is also permitted. |
|  | Instruct-XMLR$_{XL}$ | Fast food is acceptable. |
|  | XML-R$_{XL}$ | Translate this page: |

Table 19: Examples of XML-R with/without instruction tuning. MT denotes machine translation.

