# OpenReview forum: "Are Bert Family Good Instruction Followers?  A Study on Their Potential And Limitations"
_ICLR.cc/2024/Conference — ICLR 2024 poster_

### Official Review · Reviewer_bkXV · 2023-10-29

**Soundness:** 3 good
**Presentation:** 3 good
**Contribution:** 2 fair
**Rating:** 6
**Confidence:** 4

**Summary:**

We've been witnessing the rise of instruction-tuned language models since T0, FLAN, NaturalInstructions, Alpaca, etc. These models typically have a decoder-only (Alpaca) or encoder-decoder (T0, FLAN) architecture. This paper wants to study the question of whether encoder-only models such as the BERT family, can also be instruction-tuned and exhibit zero-shot task-generalization abilities.

To do so, they first develop a new MLM training scheme that mimics the Seq2Seq training paradigm by concatenating source and target sequences to feed into BERT but with proper masking so that the source sequence tokens won't attend to the target sequence tokens. When training, they mask some tokens from the target sequences to do MLM training. During inference, they first predict the length of the sequence and then iteratively predict masked tokens (all tokens are [MASK] at the beginning).

They then did instruction tuning on the 3.5B XML-R backbone model with the multilingual xP3 dataset. The performance is somewhat comparable to baselines like BLOOMZ-3B and mT0-3.7B, although still lagging behind on language generalization, generation tasks, few-shot learning, etc.


----------------------------------------------------------
POST-REBUTTAL UPDATE:
I'm raising my score from 5 to 6.

**Strengths:**

- I think the experiments are pretty solid.
- The writing is generally clear and well-organized.

**Weaknesses:**

- I'm not exactly sure what's the core contribution of the paper - it reads too much like an experiment report in this current draft. My biggest takeaway is that you can also do instruction tuning on encoder-only models. But why is that particularly "surprising"? To be fair, I'd be surprised it actually works much better than decoder-only or encoder-decoder models because then it means that the popular approach is wrong and they should go back to encoder-only models instead. But based on your experiments, it's not the case, decoder-only and encoder-decoder models still seem better at similar sizes.

- The experiments are not exactly controlled in the sense that, your baselines - BLOOMZ-3B and mT0-3.7B, are not even based on the same backbone model as Instruct-XMLR. This means that many confounders (e.g., pretraining data / steps) exist for making any scientific conclusions based on the experiments.

**Questions:**

- Why don't you also include an XMLR without instruction-tuning baseline so that we can directly see the relative improvement coming from instruction tuning?

- I am a bit confused by the method description in Section 3.2. Initially, you were describing it as if you train a separate "decoder" MLM, and H_{src} and H_{tgt} are from the two different models. But what I think you are actually doing is that, you just concatenate the source and target sequence and feed into the same BERT model, but with a modified masking scheme, such that source sequence tokens cannot attend to target sequence tokens.

---

> ### Author Response · Authors · 2023-11-23
> **Response to Reviewer bkXV**
>
> Thanks for your critical feedback, which is crucial in improving this work. We hope the following responses can answer some of your questions and mitigate your concerns.
>
> **To Question 1:** "Why don't...".
> **Response:** We agree that providing the results of XMLR without instruction-tuning can clarify whether the relative improvement comes from instruction tuning. However, it is difficult to collect the evaluation results of XMLR-w/o Instruction tuning owing to its poor performance. We provide several examples below to give an intuitive explanation (we italicize the content of the prompt).
> You can find more examples in ***Appendix A.6*** in the revised paper.
>
> We find that XML-R fails to complete these tasks without instruction tuning, i.e., it generates the special tokens [PAD] in XNLI and unrelated texts in machine translation.
>
> | | |
> |:- | :-:|
> |XNLI||
> |Source|*Suppose* "And he said, Mama, I'm home." *Can we infer that* "He didn't say a word."? |
> |Target|No|
> |Instruct-XMLR|No|
> |XML-R|[PAD]|
> |Machine translation|
> |Source|*Translate the following text from German to English:* Auch Fast Food ist erlaubt.|
> |Target|Fast food is also permitted.|
> |Instruct-XMLR|Fast food is acceptable.|
> |XML-R|Translate this page:|
> ||
>
> **To Question 2:** ''I am a bit confused..."
> **Response:** We are sorry for the confusion! Exactly, our implementation is what you have summarized "you just concatenate ...''. We also describe this in the left of Figure 1 (begin with "In practice").
>
> We describe "it as if you train ...'' at the beginning of  Section 3.2 to give a clear motivation for our concrete implementation.
> 1. A separate "decoder" is only needed if we fully follow the practice of the original encoder-decoder model. ---> 2. "This additional MLM can completely share parameters with the MLM encoder. Thus, we can adopt only one pre-training BERT model" ---> 3. The original mix-attention mechanism first acquires the last layer's source representation, and we must pass the model twice. ---> 4. We transform the original mix-attention mechanism into the dynamic mix-attention mechanism, leading to the current training format to improve training efficiency.
>
> **To Concern 1:** "What's the core contribution..."
> **Response:** We agree with you that this version merely pointed out and preliminarily analyzed some of the potential and limitations of existing Encoder-only LMs in completing various generation tasks in an instruction-following manner. There is still much work to do in the future (as another reviewer said "It can potentially lead to another series of research.'').
>
> Though preliminary, this paper still provides the following observations to the community.
>
> - It is a common impression that Decoder-only and Encoder-Decoder models are good at following instructions in generating the expected output for various tasks, including understanding, extraction, generation, etc. We demonstrate that Encoder-only also has the potential to follow instructions and complete various tasks in a generation manner.
> - The BERT family plays a vital role in the development of pre-trained LMs and are still strong competitors in many scenarios, e.g., understanding and information extraction. It is might not surprising that the BERT family can complete simple classification and understanding tasks with zero-shot prompt learning, given proper task format and subtle label name search strategy. However, no existing works provide suggestions and observations on whether the BERT family can follow instructions and complete various tasks in a generation manner. This paper fills this blank.
> - As discussed in Question 1, Encoder-only models cannot directly complete various tasks in a generation manner. We provide a basic method with details that allow for such an exploration.
> - We present the key factors to make Encoder-only models competitive with the other two variants and disclose their limitations, which might promote further study.
> - From the aspect of non-autoregressive generation, this is the first work that builds an instruction following models that can generalize across tasks and languages based on pre-trained LMs.
>
> We also agree that "It would be truly surprising ...". But we also believe the future success of any possible alternatives of existing methods/models can start from "potential", .e.g., better than mT0 on XCOPA tasks, 3.1x faster than AR models.
>
> Last, we have removed "surprisingly" from our paper.
>
> **To Concern 2:**"The experiments..."
> **Response:** We agree that "many confounders exist ...". The trends of close-source and soaring computation costs of LLMs make it difficult (sometimes impossible) to exactly control variables in experiments. To avoid any misunderstanding in our observations, we use"potential, risks, possible, etc" throughout the whole paper if necessary. On the other side, empirical observations and suggestions without fine-grained and strict control have also promoted the development of LLMs in the past year.

---

### Official Review · Reviewer_oFHF · 2023-11-01

**Soundness:** 3 good
**Presentation:** 4 excellent
**Contribution:** 3 good
**Rating:** 8
**Confidence:** 3

**Summary:**

This manuscript studies the instruction following capabilities of the BERT-family of language models, featuring an encoder-only architecture, which is one of the first of its kind. The work proposes a series of approaches that have made this possible, which eventually results in Instruct-XMLR (from XML-R), revealing promising task and language generalization abilities, previously undiscovered.

**Strengths:**

This is quite an interesting work with a refreshing perspective on language modeling via encoder-only architecture. I really appreciated the author's detailed analysis of relevant works, especially the mention of the work Wang & Cho (2019), which appears to be crucial to this work. The approaches, and training processes are detailed and well-motivated, and the benchmarks are extensive.

**Weaknesses:**

1. Some minor issues regarding citation formats (e.g. "Following the previous work Muennighoff et al. (2022), we evaluate the
model’s ability of task" is not directly readable, may need to include parenthesis)
2. As the authors have pointed out, the text generation capabilities, one of the arguably most important capabilities of language models, is still weak for Instruct-XMLR. This may limit the significance of this work.
3. Also arguably not a deal-breaker, but the compute resource (8xA100, unclear 40GB or 80GB, PCIe or SXM) and the model size (3.5B params on 0.6B tokens) may still be too small compared to cutting-edge decoder-only equivalents.

**Questions:**

It's a fascinating topic to explore beyond decoder-only autoregressive models. Would the authors agree that the benefit in text generation mainly comes from the next-token prediction formulation itself, and not necessarily the decoder architecture? One such example may be [1].

[1]: RWKV: Reinventing RNNs for the Transformer Era

---

> ### Author Response · Authors · 2023-11-23
> **Response to Reviewer oFHF**
>
> **Response to the question:**
> Thank you for acknowledging the value of this paper!
> We are not sure how much benefit the transformer decoder architecture brings. However, we do have some empirical observations about the effects of model architecture and the decoding paradigm.
> - Though the community of decoder-only structures is much larger than other architectures (owing to their merits of sample-efficiency, structure simplicity, ease of monitoring during pre-training, etc.), other architectures like transformer Encoder-Decoder (e.g., FlanT5-UL2 [1], AlexaTM [2]), RWKV [3], Convolutional Seq2Seq [4] can also achieve very promising generation performance with the next-token prediction formulation.
> - The next-token prediction (auto-regressive) decoding paradigm is one of the most effective strategies in various generation tasks, which is usually better than other paradigms, such as semi-autoregressive and non-autoregressive.
>
> We firmly believe there exist alternative architectures. There is no theoretical guarantee to prove that the existing decoder-only structure is optimal or a good sub-optimal.
>
> We think the most three import factors in self-supervised pre-training are data, model capacity, and objectives, where 1) data decides the patterns to be discovered and composed, 2) model capacity corresponds to the capability to memorize, manipulate, and generalize existing patterns, 3) the objectives correlate the learning efficiency for different patterns. In other words, the model capacity might be more important for LLMs than the structure.
>
> BTW, we also believe many valuable techniques can be started from “competitive” (like RWKV) or “potential”.
>
>
> **To Comment 1:**
> Thanks a lot for pointing out our mistakes. We have fixed the citation format issues in the latest paper version.
>
> [1] Scaling Instruction-Finetuned Language Models
> [2] AlexaTM 20B: Few-Shot Learning Using a Large-Scale Multilingual Seq2Seq Model
> [3] RWKV: Reinventing RNNs for the Transformer Era
> [4] Convolutional Sequence to Sequence Learning

---

### Official Review · Reviewer_fL5N · 2023-11-02

**Soundness:** 4 excellent
**Presentation:** 4 excellent
**Contribution:** 4 excellent
**Rating:** 6
**Confidence:** 3

**Summary:**

The authors adopt XML-R to explore the effectiveness of the BERT family for instruction following and zero-shot learning. They first design a simple yet effective strategy to utilize the encoder-only models for generation tasks and then conduct multi-task instruction tuning. Experimental results demonstrate that our fine-tuned model, Instruct-XMLR, outperforms Bloomz on all evaluation tasks and achieves comparable performance with mT0 on most tasks. Besides, Instruct-XMLR can accelerate decoding due to its non-autoregressive generation manner.

**Strengths:**

1. The idea is interesting and novel. It is great to know BERT family can also do instruction tuning. It can potentially lead to another series of research.
2. The proposed method shows competitive performance compared to auto-regressive methods. And it can achieve 3 times speed up.
3. The proposed method is simple and effective.
4. The paper is well-written and easy to read.

**Weaknesses:**

1. The method is not tested on longer sequence generation. The classification tasks don't need a longer output sequence. And machine translation fits the non-auto-regressive methods. Need to test it on the tasks like dialogue, summarization, etc.
2. It would be better to analyze how stable the method is. How do the prompt templates affect the performance. Need to report the variance or significant test.
3. The proposed method cannot handle fewshot prompt learning.

Overall, although the exploration is quite interesting, the limitations are also obvious and need further exploration. It would also be interesting to see negative results on a broader range of tasks.

**Questions:**

Have you tried to use a larger model?

---

> ### Author Response · Authors · 2023-11-23
> **Response to Reviewer fL5N**
>
> Thanks a lot for the very helpful comments in the first round of discussion! We hope the following responses can answer some of your questions.
>
>
> **To Comment 1:** "Longer sequence generation tasks like dialogue, summarization, etc."
> **Response:** Exactly! It is essential to discuss the performance of the introduced method on longer sequence generation tasks.
> Besides machine translation, *we further provide evaluation results on **5** additional sequence generation tasks with **6** datasets* (with longer output sequences than most previously reported tasks).  We summarize these tasks and datasets below. *More details of these tasks and datasets can be found in **Appendix A.4*** of our revised paper.
> | Tasks     |      Datasets            |
> | - | - |
> | Text Summarization     | XSUM, Gigaword |
> | Text Simplification        | WIKI-AUTO |
> | Paraphrase Generation    | QQP |
> | Controlable Generation | COMMONGEN|
> | Dialogue Generation          | PersonaChat |
> |  |  |
>
> We present the experimental results below, along with ***preliminary analysis***. We give the variance (this number denotes the performance range, i.e., the performance boundary) of ***5 different prompts*** in the bracket. Since length prediction plays a key role in language generation tasks in non-autoregressive models, we also report the results of Instruct-XMLR decoded ***with ground truth length (w/ GL)*** to explore the effects of length prediction.
>
> R-1/2: ROUGE-1/2, B-2: BLEU-2, D-2: Distinct-2
> | Dataset|XSUM |Gigaword|WIKI-AUTO|QQP|COMMONGEN|PersonaChat|
> |:-:|:-:|:-:|:-:|:-:|:-:|:-:|
> |Score| R-1/R-L| R-1/R-L| B-2/D-2|B-2/D-2|B-2/D-2| B-2/D-2|
> |BLOOMZ-3B | 35.12/27.97| 31.45/28.94 |41.90/0.57|38.24/0.54|35.20/0.44|16.39/0.39|
> |mT0-3.7B| **37.85**/**30.28**| **34.31**/**31.54** | 55.92/0.64|51.93/0.63| 31.86/0.36|9.17/0.39|
> |Instruct-XMLR-XL|35.62/27.90|30.64/28.52|47.91/0.65|43.50/0.67|21.68/**0.52**|2.88/0.52|
> |Instruct-XMLR-XL w/ GL| 36.00/27.84|32.60/29.85|**56.93**/**0.76**|**60.02**/**0.69**|**35.44**/0.48|**17.30**/**0.53**|
> | | | | ||||
>
> In general, we can find that:
> - The results in most tasks are consistent with what we mentioned in the original paper, i.e.,  Instruct-XMLR-XL outperforms BLOOMZ-3B on 4 out of 6 tasks (with the left 2 tasks comparable) and underperforms mT0-3.7B without target length.
> - Length prediction significantly impacts some specific tasks where there is no strong alignment relationship between source length and target length (e.g., COMMONGEN and PersonaChat). Since the BLEU score is sensitive to length, length prediction leads to a serious decline compared to decoding with ground truth length.
> - Instruct-XMLR can generate more diverse texts for all text generation tasks due to the non-autoregressive generation manner, i.e., unlikely to be trapped by common n-gram patterns.
>
> **To Comment 2:** "To analyze how stable the method is. Need to report the variance or significant test."
> **Response:** We are sorry to miss it in the previous version. In our revised PDF, we supplement the difference of various prompts and the correlated analysis in ***Appendix A.3***.
>
> **To Comment 3:** "The proposed method cannot handle fewshot prompt learning."
> **Response:** We have tried to make up for this flaw by including some few-shot instruction data for training rather than the only zero-shot instruction data we adopted in our paper. Unfortunately, this doesn't bring any profit. Part of the results are given below, where more details can be seen in ***Appendix A.5***.
> " ***w/zs** denotes the model trained on zero-shot instruction data, **w/fs** denotes the model trained on mixed zero/few-shot instruction data, **w/ICL4** denotes we adopt four examples in the demonstration for few-shot prompt learning.* ''
> - Results for seen languages.
>   | Model|XCOPA|XNLI|XWinograd|
>   |:-|:-:|:-:|:-:|
>   | Instruct-XMLR w/zs|0.646|0.518|0.582 |
>   | Instruct-XMLR w/zs w/ICL4 | 0.548 | 0.414 | 0.547 |
>   | Instruct-XMLR w/fs | 0.642 | 0.489 | 0.565 |
>   | Instruct-XMLR w/fs w/ICL4 | 0.571 | 0.442 | 0.553 |
>   ||||
>
> **To Question:**
> We are now experimenting with the biggest version of the BERT Family, the 11B XMLR-XXL model, with only around 1/2
> data of the original Instruct-XMLR-XL model being processed till now. We present the intermediate results below to give intuitive analysis.
>
> - Results for unseen languages.
>   | Model  |  XCOPA | XNLI | XWinograd |
>   |:-|:-:|:-:|:-:|
>   | Instruct-XMLR-XL | 0.599 | 0.466 | 0.539 |
>   | Instruct-XMLR-XXL| 0.593 | 0.523 | 0.562 |
>   ||||
>
> We can find that:
> - With only 1/2 instruction data, Instruct-XMLR-XXL already outperforms Instruct-XMLR-XL on XNLI and XWinograd, indicating the positive effects of larger model size.
> - Compared with Instruct-XMLR-XL, Instruct-XMLR-XXL seems more effective for language generalization.

---

### Meta-Review · Area_Chair_t6Kq · 2023-12-02

**Metareview:**

This paper studies whether encoder-only models such as the BERT family, can also be instruction-tuned and exhibit zero-shot task-generalization abilities. After rebuttal, it received scores of 668.

On the one hand, the reviewer who gave a score of 8 is excited about this paper, commenting that this is quite an interesting work with a refreshing perspective on language modeling via encoder-only architecture. Other reviewers also agree that the experiments are solid, and the writing is generally clear and well-organized. On the other hand, based on the experiments, encoder-only models still lag behind in decoder-only and encoder-decoder models in instruction following and cannot perform well in few-shot prompting, and the model needs to be tested more extensively on longer-form text generation.

Overall, all the reviewers are positive about the paper, the AC would like to recommend acceptance of the paper.

**Justification For Why Not Higher Score:**

Although the exploration is interesting, the limitations are also obvious and need further exploration. The text generation capabilities are still weak for Instruct-XMLR, which may limit the significance of this work.

**Justification For Why Not Lower Score:**

All the reviewers are positive about the paper, it deserves to be accepted.

---

### Decision · Program_Chairs · 2024-01-16

Accept (poster)